### 1 Formation and dynamics of sandy dunes in the inland areas of the Hexi Corridor

- 2
- 3 Bing-Qi Zhu
- 4 Key Laboratory of Water Cycle and Related Land Surface Processes, Institute of Geographic Sciences and
- 5 Natural Resources Research, CAS, Beijing 100101, China
- 6
- 7 Abstract:
- 8

9 Dynamic changes of aeolian landforms and desertification under global warming in a middle-latitude 10 desert belt, the Hexi Corridor in China, considered to be one of the source and engine area of sandstorms in China and Northern Hemisphere (NH), is a typical problem of climate change and landscape response, 11 which need a comprehensive understanding of the history and forcing mechanisms of recent landform and 12 13 environmental changes in the region. Based on the existing high-resolution satellite image interpretations, field investigations and observations, comprehensive evidences from geomorphological, aeolian-physical, 14 granulometrical and geochemical analysis, this study discussed the formation of dune landforms, the 15 mechanism of desertification and their environmental implications in the Hexi Corridor. The analytical 16 results show that 80% of the sand particles flow within a height of 20~30 cm near the surface, and about 17 18 half of the sand particles flow within a height of 0.3~0.5 cm near the surface in the Hexi Corridor. The 19 average height of the typical crescent-shaped dunes is about 6.75m, and the minimum and maximum values are between 2.6 and 11.2m. On the inter-annual and multi-year time scales, only the 20 crescent-shaped dunes and chains of barchan dunes are moving or wigwagging in the study area, while the 21 parabolic and longitudinal dunes did not move. Under the influence of wind speed, strong wind days and 22 other factors, the dunes at the edge of the Mingin Oasis move the fastest, with a moving speed of about 23 6.2m/a. Affected by the main wind direction and other factors, the dunes at the edge of the Dunhuang 24 Oasis move the slowest, with a moving speed of about 0.8m/a. The main factors affecting the dynamic 25 changes of sandy dunes in the Hexi Corridor are the annual precipitation, the annual average wind speed 26 27 and the number of annual strong wind days, of which the annual precipitation contributes the largest, indicating that the climate factors have a most important impact on the dynamic change of sand dunes. 28 29 The cumulative curve of particle size frequency of dune sediments in the Hexi Corridor basically presents a 30 three-segment model, indicating a saltation mode dominated under the action of wind, but superimposed with a small amount of coarser and finer particles dominated by the creeping and suspension models, 31 which is obviously different from that of the Gobi sediments with a dominant two-segment mode. The 32 palaeo-geographical, sedimentological and geochemical evidences indicate that dune sediments in the 33 Hexi Corridor are mainly derived from "locally or in-situ raised sandy sediments", which are mainly come 34 from alluvial plains and ancient fluvial sediments, as well as ancient lake plains and lacustrine deposits, 35 1

aeolian deposits in the piedmont denudation zones of the north and south mountains and modern fluvial sediments in the corridor. In geochemical compositions of major and trace elements, the dunes in the Hexi 37 Corridor have certain similarities and differences to other sandy dunes in the northwest and northern 38 39 deserts of China or aeolian loess in the Loess Plateau. Sandy dunes in the Hexi Corridor are relatively rich in iron and Co. Considering the proportion of fine particles on the surface, the coverage rate of surface salt 40 crust, and the potential migration of erodible sandy materials, it can be concluded that the Gobi area in 41 42 the west Hexi Corridor is not the main source area of sandstorms in the middle and east of the corridor, but the north probably is. In the past half century, the warming and humidification of local climate is the 43 main cause of the reduction of sandstorms in the study area, and the Hexi Corridor has a potential trend of 44 anti-desertification, which is mainly controlled by climate change but not human activities. For the oasis 45 areas of the corridor, however, the effective measures to restrict desertification depend on human 46 activities. Restriction of the decline of groundwater is the key to preventing desertification in oases, rather 47 than water transfer from outer river basins. 48

### 51 Keywords:

Sandy dunes; Geomorphology; Sedimentology; Geochemistry; Desertification; Hexi Corridor.

### 54 1. Introduction

Aeolian sediments and their sedimentary strata record the environmental changes in the source areas 56 or desert areas and their responses to global climate change and human activities. They are unique and 57 important sedimentary and geomorphological archives of dryland landscape evolution (Goudie, 2002; 58 Lancaster et al. ., 2013, 2016; Williams, 2014; Yang et al., 2019). In China, about 566,000 square kilometers 59 of land area are covered by aeolian sand, covering a wide range of geomorphological and tectonic 60 backgrounds ranging from 155m below sea level to 5,000m above sea level (Yang, 2006). The desert 61 landscapes dominated by active sandy dunes are mainly distributed in the arid areas with an average 62 annual precipitation of less than 200 mm, while the sandy-land landscapes dominated by semi-active 63 dunes and vegetated dunes mainly appeared in the semi-arid areas with an average annual precipitation of 64 65 200-400 mm (Zhu et al., 1980). The present geomorphology of these sandy deserts is the product of long-term and short-term changes of the interaction between endogenic forces (such as tectonic 66 movement) and external forces (such as climate) of the earth system. In turn, these deserts may directly 67 affect the global climate system through sediment circulation (such as dust cycles) (Yang, 2006). Therefore, 68 the understanding of desert landscape evolution will increase our understanding of the earth system. 69

Regarding the formation and evolution of desert landscapes in China, the loess-paleosol sedimentary

sequences from the Loess Plateau indicate that the deserts in northwest China may have existed as early as 22 myr (Guo et al., 2002, 2004), but the geomorphological and sedimentological evidences found inside 72 these deserts indicate that the modern-scale landscapes of these deserts are much younger in age (Yang et 73 74 al., 2004, 2006). But up to now, due to the lack of long-enough and continuous stratigraphic profiles, the geomorphological connection between the Tertiary desert and the present desert is still unclear. And in 75 many areas of the deserts in northwest China, lacustrine and fluvial sediments from the Late Pleistocene 76 77 and even Holocene are buried under the sandy dunes, which indicate that the environment of these desert areas has changed dramatically during the late Quaternary (Yang, 2006; Chen et al., 2020). For example, in 78 79 the Badanjilin Desert near the northeast of the Hexi Corridor in China, although the formation mechanism 80 of the giant sandy dunes in the desert is still a dispute in people's understanding, such as "the theory of climate control", "the theory of tectonical/geomorphological control" and "the theory of groundwater 81 control", geomorphological survey is essential to resolve this dispute. It is conceivable that the dynamic 82 83 genesis of sand dunes, namely desertification, will be crucial for understanding this problem, where the dune landforms and desertification processes cover almost all the important information archives that 84 understanding the earth system. 85

The movement of aeolian materials and related formation, dynamics and evolution of dune landforms 86 are the results of the transportation and accumulation of sandy sediments under the influence of climate 87 88 (especially wind and atmospheric circulation), which is the direct cause of landsurface desertification and 89 one of its important manifestations (Zhu et al., 1980; Zhu and Wang, 1992; Yang et al., 2004, 2019). For example, the ruins of ancient cities are buried by shifting sands in northern China and some famous 90 steppes in history but are occupied at present by desert landscapes with undulating sandy dunes, which 91 are clear evidences of land desertification in the past 2 Ka (Zhu and Wang, 1992). The movement of sand 92 dunes not only affects the development and safety of agriculture and transportation, but also reflects the 93 modern geomorphological processes of landform development in arid areas and its environmental 94 response to global changes. Therefore, it is of great significance to study the formation and dynamic 95 characteristics of various dune landforms in different regions of the world to reveal desertification and 96 97 environmental changes in drylands.

The formation and dynamics of sandy dunes in the world were observed and studied for the first time 98 99 in the United States (Finkel, 1959) and the former Soviet Union (Znamenski, 1962) in the 1950s. During this 100 period, the famous desert physicist Bagnold put forward the formation mechanism of mobile sandy dunes and the formula of moving velocity of active dunes (Bagnold, 1959). Dune formation and dynamics in 101 102 China were qualitatively or semi-quantitatively described in most early studies (Yang, 2006). For example, some pioneer scholars have studied the development and movement of sandy dunes in the Taklamakan 103 Desert, and they quantitatively analyzed the moving speed and evolution process of local crescent sandy 104 dunes (Zhu et al., 1964; Zhu et al., 1980, 1981). However, these studies still laid a solid foundation for the 105

later development of refined and quantitative researches due to the progress of research methods andtechnical tools, and they are still a milestone cornerstone of desert researches in China.

The Hexi Corridor in northwestern China at the middle-latitudes of Northern Hemisphere (NH) was 108 109 once one of the most important trunk sections of the world-famous Silk Road, and also a place where several ancient cultures converged. However, today it is facing severe problems of desertification and 110 climate change under global warming. For nearly half a century, frequent sandstorms in northern China 111 have been considered to be the notorious tragedy and direct consequence of the desertification in the 112 Hexi Corridor, because the Hexi area is considered to be the main source area and the engine area of 113 sandstorms in China (Zhang and Ren, 2003; Pu, 2005; Li and Zhang, 2007). Therefore, the problem of 114 desertification in the Hexi Corridor is one of the major problems that have been urgently needed to be 115 resolved in Gansu Province and even in northern China for half a century. 116

The purpose of this study is, based on the comprehensive evidences from the extensive dune geomorphological survey, the sedimentological and geochemical analysis of dune sediments, and the meteorological analysis of local weather records in the past several decades, to understand the genesis and dynamic changes of sandy dunes in the Hexi Corridor and its relationship with climate change during the past half century, and to explore the mechanism of local desertification in the Hexi Corridor and its environmental implications.

123

### 124 2. Background and analytical methods

# 126 2. 1. Geographical, geological, geomorphological and hydrological backgrounds of the Hexi Corridor 127

The Hexi Corridor is located in the central and western parts of Gansu Province in Northwest China 128 129 (Fig.1), including Wuwei, Jinchang, Zhangye, Jiuquan, Jiayuguan and other cities in the west of the Yellow River, with a total area of approximately 5,100 square kilometers. In terms of regional geomorphology, the 130 Hexi Corridor is located in the lowland area between the Qilian Mountains and the Alashan Plateau. The 131 Alashan Plateau in the north of the Hexi Corridor distributes three large sandy deserts of China, i.e. the 132 Badanjilin Desert, the Tenggeli Desert and the Ulanbuhe Desert. For the Qilian Mountains in the south of 133 the Hexi Corridor, the melting water of ice and snow in the Qilian Mountains in the south converges into 134 135 several large rivers flowing northward into the Hexi Corridor, such as the Heihe River, the Shiyang River and the Shule River, etc. In the middle and lower reaches of these rivers flowing through the south corridor, 136 diluvial and alluvial fans are well developed, and hydrologically, they are also the main locations of spring 137 overflow zone of each catchment derived from the Qilan Mountains. Oases are widely developed in the 138 toes of these alluvial fans and are the major agricultural exploitation areas and the resident agglomeration 139 areas of northwest China. 140

In climate, the Hexi Corridor is located in the center of temperate desert belt in the mid-latitudes of 142 Northern Hemisphere. Except for the forest and grasslands distributed in the middle- and high-elevation 143 mountain areas in the south, most of the Hexi Corridor is under a typical arid desert climate with desert 144 landforms developed. The desert types are dominated by Gobi desert and sandy desert, which account for 145 46.64% of the total area of the region.

In history, the Hexi Corridor was a necessary place for the famous ancient Silk Road in China. In modern times, however, the expansion of population and socio-economic development of the Hexi Corridor, as well as the human-caused competitive redistribution of water resources, have led to the onset and enhancement of desertification in the Hexi Corridor (Pu, 2005). The expansions of sandy dunes and dune fields in the corridor, and even the combination with surrounding sandy desert, have occurred in the past 2 ka (Zhu and Wang, 1992; Ren et al., 2014).

In terms of aeolian landforms, sands dunes or dune fields in the Hexi Corridor are mainly distributed 152 in a narrow and long belt between the Qilian Mountains and the Heli mountains to the west of Wushaoling 153 and the east of Palaeo-Yumenguan (Fig. 2) (Zhu et al., 1980). Compared with the dune landforms in the 154 adjacent areas of the Badanjilin and Tenggeli Deserts, the sandy dunes in the two deserts tend to be 155 convergent in spatial distribution, while the Hexi Corridor is different, where the dunes are almost 156 scattered, mainly distributed in the vicinity of oases along some rivers, in the oasis or Gobi desert areas 157 158 (Fig. 2). From east to west in the dune belt, sand dunes are mainly distributed around the Mingin Oasis in 159 the lower reaches of the Shiyang River, the Zhangye and Gaotai oases in the middle reaches of Heihe River, the Jiuquan and Jinta oases in the lower reaches of the Beidahe River, and the Dunhuang oasis in the lower 160 reaches of the Danghe River (Fig. 2) (Zhu et al., 1980). 161

The total area of dune fields in the Hexi Corridor is about 754 square kilometers, and a large number of big crescent-shaped dunes and chains of crescent-shaped dunes develop on the edges of oases. The Minqin Basin is a typical area with dune landforms development in the Hexi Corridor. It is located at the lower reaches of the Shiyang River and the western edge of the Tenggeli Desert. The annual average precipitation is about 116.4 mm and the annual average wind speed is about 2.25 m/s. A large number of crescent-shaped sandy dunes are distributed on the northwestern edge of the oasis, i.e., the windward of sand-transport winds in the oasis.

### 170 2. 2. Data and analytical methods

For the study of sandy dune geomorphology, the first method is to use the sample-quadrate survey procedure to measure the height and shape of typical high dunes in the field with a rangefinder, and the second is to measure the length, angle and width of the windward slope and downwind slope of each dune in the sample quadrate and between different quadrates along the local dominant wind direction by

176 using rangefinder and remote sensing image scales (such as Google Earth scales, etc.), and then the comprehensive geomorphic data of sandy dunes in the region is obtained. In addition to the 177 geomorphological data of sandy dunes themselves, landscape researchers will also use the 178 179 sample-quadrate survey method to investigate the ecological parameters of vegetation cover in the selected sampling area. For both geomorphological and ecological surveys, sub-scale sample quadrates will 180 be selected from the upper, middle and lower parts of the windward and leeward slopes of each dune. 181 182 Three quadrates can be selected from the dune slope in the windward and downwind directions of each dune along the local prevailing wind direction and the size of each quadrate can be designed as 5m × 5m 183 or smaller (Chang et al., 2016a, 2017; Lang et al., 2017). 184

In recent years, a number of studies have been systematically carried out in different areas of the Hexi Corridor to investigate the different landform types of widespread sandy dunes at a geomorphic unit scale 186 in the field, including the crescent-shaped (barchan) dunes, chains of barchan dunes, pyramid-shaped 187 188 dunes, parabolic dunes and longitudinal dunes belt. Based on field observations and satellite remote sensing image data in different periods, the geomorphological parameters and characteristics of these 189 dunes are obtained (Zhang and Dong, 2014; Chang et al., 2016a, 2017; Lang et al., 2017). The 190 geomorphological parameters of part of these dunes in the Hexi Corridor and their comprehensive data 191 are shown in Tables 1 and 2. 192

In addition to the above-mentioned intuitive survey and measurement of geomorphic parameter of 194 sandy dunes, quantifying the structure of wind-blown sand flow and the movement rate of dunes is also the most direct and effective means to explain the dynamic change of dunes and their geomorphological 195 evolution (Dong et al., 1998; Chen and Liu, 2011; He et al., 2012; Dong and Huang, 2013; Wang et al., 2013; 196 Hu et al., 2016; Mao et al., 2016). Generally, there are two methods to study the moving velocity of sandy 197 dunes, one is early positioning observation (MDCES, 1975; Dong et al., 1998; He et al., 2012) and the other 198 199 is based on remote sensing images (Chen and Liu, 2011; Dong and Huang, 2013; Mao et al., 2016). Research works based on the both methods have been carried out in dune fields of the Hexi Corridor. On 200 this basis, this study will integrate and organize the different observation data of dune movement 201 measurement in the Hexi Corridor, and further discuss the geomorphological evolution of sandy dunes in 202 203 the Hexi corridor.

The grain size composition and distribution of aeolian sediment is an important indicator to understand the formation and development of sand dunes. This is because the grain size parameters of sand particles can be used not only to distinguish the depositional environment (aeolian, fluvial or lacustrine), but also to identify the movement types (creep, saltation or suspension) of sediments in the transportation process. Therefore, the analysis and study on the granular sedimentology of sandy dunes is a basic method to understand the genesis and evolution of the dunes in the Hexi Corridor. At present, research works about the grain-size sedimentology of aeolian sediments and related aqueous sediments,

such as alluvial and proluvial fans, lacustrine deposits, fluvial deposits, has been widely carried out in the Hexi Corridor (Zhu and Yu, 2014; Zhu et al., 2014; Zhang and Dong, 2015; Zhang et al., 2016; Pan et al., 2019; Zhang et al., 2020). On this basis, this study systematically collects and organizes the granular evidences, which makes it possible to conduct a comprehensive and comparative study on the dunes in the Hexi Corridor from a perspective of sedimentology.

Erodible clastic sediments as the material sources are the fundamental base for the formation of 217 sedimentary landforms (Pettijohn et al., 1972; Taylor and McLennan, 1985). Therefore, identifying the source of wind-induced materials in an arid environment is a prerequisite for understanding the formation 218 219 of dune landforms (Zhu et al., 1980, 1981; Yang et al., 2012). The analysis of major and trace elements, 220 including rare earth elements, has become a reliable technique for detecting the source of desert sediments (Muhs et al., 1995, 1996; Pease et al., 1998; Honda and Shimizu, 1998; Wolfe et al., 2000; Pease 221 and Tchakerian, 2003; Zimbelman and Williams, 2002; Muhs, 2004; Yang et al., 2007; Zhu and Yang, 2009; 222 223 Jiang and Yang, 2019). The reason is that for aeolian sediments, the differences in compositions and distributions of rare earth elements and other trace elements in different samples/sub-fractions are largely 224 controlled by the parent-rock compositions, because these elements only exist in specific minerals and are 225 difficult to be lost during transportation (Pettijohn et al., 1972; Taylor and McLennan, 1985). In the Hexi 226 Corridor, preliminary results have been achieved in the case studies of analyzing the elemental 227 228 compositions of aeolian sediments using major- and trace-element geochemical methods (e.g., Ren et al., 229 2014; Pan et al., 2019; Zhang et al., 2020), which provide basic data for this study to comprehensively identify the material sources of different dunes in the study area. 230

The continuous data records of different meteorological parameters of local weather stations in the 231 Hexi Corridor in the past half century, such as temperature, precipitation, relative humidity, wind speed, 232 strong wind days and sandstorms days, will not only be the basis for this study to discuss the regional 233 climate change under the background of global warming, but also the basis for exploring the response of 234 regional landscape to climate change based on the statistical relationship between geomorphic parameters 235 and climate parameters on a multi-decade time scale. Therefore, this study will collect and use the 236 237 meteorological data of the Hexi Corridor for nearly half a century to analyze the regional climate change and its relationship with the dynamic changes of dune landforms. 238

3. Results

3. 1. Geomorphological parameters (Height, shape, and dynamics) of sandy dunes in the Hexi Corridor

The comprehensive data on the heights of different types of sandy dunes widely developed in different areas of the Hexi Corridor, as well as other geomorphic parameters of these dunes, can be found

in Table 1, Table 2 and Fig. 3. It can be seen from Table 1, Table 2 and Fig. 3 that the average height of typical crescent-shape (barchan) dunes in the Hexi Corridor is about 6.75m, the maximum is about 11.20m, 247 and the minimum is only about 2.60 m. The average height of typical chains of barchan dunes in the study 248 249 area is about 9.23 m, the maximum is about 13.80 m, and the minimum is only about 5.80 m. The typical pyramid-shaped dunes in the study area have an average height of about 86.25 m, with a maximum of 250 about 121.80 m and a minimum of about 25.80 m. The average height of typical parabolic dunes in the 251 252 study area is about 4.08 m, the maximum is about 4.60 m, and the minimum is only about 3.38 m. The average height of typical longitudinal dunes in the study area is about 13.02 m, the maximum is about 253 254 18.60 m, and the minimum is only about 5.60 m.

Regarding the dynamic changes of sandy dune landforms, as early as 1959-1964, the newly established Mingin Comprehensive Experimental Station of Desertification Control (MCESDC) in China 256 carried out the field positioning observation and research on wind-blown sand flows in the Hexi Corridor 257 258 (Zhu et al., 1980; Zhu, 1994, 1999; Zhu and Wang, 1992; Wang, 2003). For example, 187 positioning observation points were set up in field along a 20km long observation line in the Minqin Basin and a large 259 number of observations were made to assess the structure of wind-blown sand flow, the shape of sand 260 dunes, erosion and accumulation of aeolian sand, changes in sand ripples and sand dune movement, etc. 261 Later works continued to carry out relevant researches on different areas of the Hexi Corridor (MDCES, 262 263 1975; Zhu et al., 1980; Wang, 2003; Zhang et al., 2004; Qu et al., 2005; Wang et al., 2013; Yin et al., 2014, 264 2016; Chang et al., 2016a, 2017; Zhang et al., 2016; An et al., 2019; Chang, 2019; Hu et al., 2020). Results of these studies show that in the Hexi Corridor, 80% of the sand particles of the wind-blown sand flows are 265 moving in the height of 20 ~ 30cm near the ground surface, of which about half of the sand particles are 266 moving in the height of 0.3 ~ 0.5cm near the surface. At the wind speed of 7m/s, 75% of the sand particles 267 are within the height of 10cm, and only 0.035% are within the height of 76 ~ 200cm. The movement 268 modes of sand particles in the Hexi Corridor include creeping (wriggling/rolling), saltation 269 (jumping/springing) and suspension (floating/levitating), and the movement modes of sandy dunes 270 include three ways: straight-forward movement, wigwagging movement, and onward-wigwagging 271 272 movement (Wang, 2003).

Among the different sandy dunes in the Hexi Corridor, the crescent-shaped (barchan) dunes, chains of barchan dunes, pyramid-shaped dunes, parabolic dunes and longitudinal dunes have received the most extensive attention. Based on field surveys/measurements (e.g. measurement of the electronic total station) and satellite image (e.g. Google Earth) data of different periods, the researchers obtained the geomorphological parameters and moving speeds of these different dunes (e.g. Ren et al., 2010; Chang et al., 2016a), as shown in Table 1 and Fig. 3.

It can be seen that the average moving speed of the crescent-shaped dunes is about 6.62 m/a, the maximum is 12.51 m/a, and the minimum is only 1.01 m/a (Fig. 3a). The average moving speed of the

281 chains of barchan dunes is about 6.54 m/a, the maximum is 8.30 m/a, and the minimum is only 5.34 m/a (Fig. 3b). Compared with the crescent-shaped dunes, the chains of barchan dunes move relatively slowly 282 and the movement speed changes little. In general, the crescent-shaped dunes and the chains of barchan 283 284 dunes in the Hexi Corridor move along the NW-SE direction. The direction of movement of the east part of the corridor is about N45° W, while the movement angle of the Jinta area in the western corridor increases 285 (Table 1 and Fig. 3). The average swing velocity at the tops of the pyramid dunes is about 6.32 m/a, the 286 maximum is 97.37 m/a, and the minimum is only 1.14 m/a. The direction of movement of the pyramid 287 288 dunes will also change, but its main direction of motion is SW-NE (Fig. 3).

### 290 3. 2. Granular sedimentology of sandy dunes in the Hexi Corridor

In the Jiuquan Gaotai area in the middle and Eastern Hexi Corridor (area B in Fig. 2), grain size parameters (including mean, standard deviation, skewness and kurtosis, etc.) of sand dunes at different 293 294 geomorphological positions on the dune surface, such as the toes of windward slope, slope surface, dune 295 crest (top), toes of leeward slope, were determined (Zhang and Dong, 2015). The granular sedimentology shows that the grain-size frequency cumulative curves of sand dunes in this area are mostly unimodal, and 296 297 a few are bimodal (Fig. 4); the dune surface sediments are mostly fine sand fraction and very fine sand 298 fraction, with an average grain size of 0.07 mm±0.01~0.24 mm±0.06, which is similar to the average particle size of sand dunes in the world. The finer the dune particle is, the better the sorting degree is. The 299 300 mean grain size increases with the increase of the skewness values, but decreases with the increase of the kurtosis values. From upwind to downwind, the dune sediment becomes finer, the medium-sand fraction 301 302 decreases, and the fine-sand fraction, very-fine-sand fraction, silt-fraction and clay-fraction increase; the 303 source materials affect the changes in the average grain size of dune sediment from upwind to downwind (Zhang and Dong, 2015). In this dune field, there are three types of grain-size-distribution patterns in a 304 dune-scale unit: the dune crest is coarser (the dune slope and inter-dune area are finer), the dune crest is 305 306 finer (the dune slope and inter-dune land are coarser), and there is no significant difference between dune crest, windward and leeward slopes. Among them, the coarser-dune-crest model is the most common type, 307 308 accounting for 69% of all sandy dunes, while the finer-dune-crest is the second most common type 309 (accounting for 24%) (Zhang and Dong, 2015).

In the Jinta-Jiayuguan-Huahai area in the western Hexi Corridor (area A in Fig. 2), the crescent-shaped (barchan) dunes developed on the Gobi desert and ancient playas have been systematically studied on granular sedimentology (Pan et al., 2019). The results of this research show that the grain size of the surface aeolian sediments of crescent-shaped dunes in the western Hexi Corridor is mainly the medium-sand fraction ( $21.7 \sim 57.4\%$ ), followed by the fine-sand fraction ( $23.2 \sim 53.0\%$ ); the mean grain

size ranges between  $0.27 \sim 0.43$  mm (while the paleolacustrine sediment ranges between  $0.10 \sim 0.21$  mm) (Pan et al., 2019). The crescent-shaped dune sediments in this region are mainly medium to good in the sorting level. The frequency cumulative curves of dunes are mostly unimodal and nearly symmetrical, and the kurtosis is medium in level. The granular characteristics of sand dunes in this region are closely linked to their dune morphology and the properties of the underlying Gobi surface (Pan et al., 2019).

### 321 **3. 3. Geochemical and sources of dune sediments in the Hexi Corridor**

Regarding the source of aeolian sediments, the provenance of sandy dunes in the Hexi Corridor was firstly investigated as early as 1959-1964 by the China's Minqin Comprehensive Experimental Station of Desertification Control (MCESDC) (MDCES, 1975; Zhu et al., 1980; Chang, 2019), and continues to this day (Ferrat et al., 2011; Ren and Wang, 2010; Ren et al., 2014; Zhang and Dong, 2015; Zhang et al., 2016, 2020; Chang, 2019).

Due to the application of geochemical methods in recent years, it is possible to identify sediment sources more precisely (Wang, 2011; Wang and Wang, 2013; Ren et al., 2014; Pan et al., 2019; Zhang et al., 2020). In this study, based on geochemical evidences, we take the Gobi areas in the Huahai-Jiayuguan-Jinta region of the west Hexi Corridor (area A in Fig. 2), the Jinta-Gaotai region of the middle Hexi Corridor (area B in Fig. 2) and the Minqin Basin in the east Hexi Corridor (area C in Fig. 2) as the case examples (Ren et al., 2014; Pan et al., 2019; Zhang et al., 2020) to explore the material sources of sandy dunes developed in these regions.

The Minqin Basin is dominated by an oasis landscape. It is located in the east Hexi Corridor (area C in Fig. 2), and the northern edge of the Loess Plateau, and is bordered by the Badanjilin Desert to the northwest and the Tenggeli Desert to the southeast (Figs. 1 and 2). The Minqin Basin is considered a natural obstacle to the convergence of the two deserts (Zhu et al., 1980). Geographically and geomorphologically, identifying the origin and transportation of aeolian sediments in the Minqin oasis and its adjacent desert areas will help to better understand the relationship between loess and desert in China (Liu, 1985; Sun, 2002; Yang et al., 2007a; Yang et al., 2011; Ren et al., 2014).

The research works of Ren et al., (2014) and others (Ren, 2010; Ren and Wang, 2010) systematically collected aeolian sediment samples from sandy dunes in the Minqin Oasis and its surrounding desert areas. Through geochemical analysis, combined with wind data and cluster analysis methods, the characteristics of compositions and spatial distributions of major and trace elements of aeolian samples from the Minqin Oasis and its adjacent deserts (the Badanjilin and Tenggeli Deserts), as well as the provenance and transportation pathways of aeolian sediments in these areas, were discussed. The analysis of geochemical data shows that in the bulk (whole-rock) samples of sandy dunes in the Minqin Basin (M) and its

surrounding areas (the Badanjilin Desert B, the Badanjilin-Minqin transition zone BM, the Tenggeli-Minqin transition zone TM, the northeast edge of the Tenggeli Desert TNE, the southwest edge of the Tenggeli 350 Desert TSW), the contents of major elements are higher in the content of SiO<sub>2</sub>, reaching between 72.2% 351 352 and 88.9% with an average of about 83.3%. In contrast, the contents of most trace elements are relatively low, and only the contents of Ba, Ce, Co, Mn and Sr reach > 100 ppm (Ren, 2010; Ren et al., 2014). 353 Compared with the average composition of the upper continental crust (UCC, Taylor and McLennan, 1985), 354 355 the concentrations of Ba,  $SiO_2$ , Rb, Sr,  $Al_2O_3$  and  $K_2O$  in the Minqin Basin and its surrounding areas are relatively uniform (Fig. 5), indicating that the spatial differences of these elements in abundance are small 356 357 and they are relatively homogeneous in the study area, while obvious convex and concave shapes are 358 observed for other elements (Fig. 5), indicating that the spatial differences of these elements in abundances are large and they are relatively heterogeneous in the study area. The homogeneous and 359 heterogeneous characteristics between different elements thus can be used as geochemical indicators to 360 361 identify different sources of sediments in the study area. For the major elements' compositions, only SiO<sub>2</sub> is enriched relative to UCC, and the others are relatively depleted (Fig. 5). For the trace elements' 362 abundance, most elements are depleted, except for Cr and Ni enriched in B and BM area and Cr enriched 363 in TNE area. The binary and ternary diagrams of some major and trace elements and their ratios, such as Cr, 364 Ni, Cr/V, Y/Ni, Al, V, Zr, Hf, Zr/Hf, reveal that sandy dunes have different material sources between the 365 366 western part of the Minqin Basin (including sub-area B, BM and TNE) and the southeast side of the Minqin 367 Basin (TSW), while sand dunes in the Mingin Basin (M) and the Tenggeli-Mingin transition zone (TM) are related to the two big deserts, respectively (Ren et al., 2014). 368

Some researchers have conducted geochemical analysis of major and trace elements in aeolian 369 sediments of barchans dunes and other sediments developed in the western Hexi Corridor (Zhang et al., 370 2017; Pan et al., 2019). The dunes studied are located in the Gobi area to the north and west of Jiayuguan 371 (area A in Fig. 2). The dune types are mainly barchan dunes, chains of barchan dune and asymmetric 372 barchans dunes (Zhang et al., 2017; Pan et al., 2019). The aeolian samples were mainly collected from the 373 surface sediments of barchan dunes and asymmetric barchan dunes, including different geomorphic sites 374 375 of dunes such as the crest of dune, the bottom of the windward slope, the middle of the windward slope, and the bottom of the leeward slope. The analytical results show that after the standardization of UCC, the 376 barchan dunes on the Gobi surface in the western Hexi Corridor are significantly enriched in the major 377 378 elements CaO and SiO<sub>2</sub> (accounting for 5.55% and 66.12% of the total rock mass, respectively). The element contents of Cao, MgO and  $Fe_2O_3$  are gradually enriched from northwest to Southeast, that is, the 379 enrichment degree increases along the dominant wind direction. The UCC-normalized concentrations of 380  $Na_2O$  and  $K_2O$  are both significantly 

384 variations of trace elements are similar in different geomorphic positions of the one dune (Pan et al., 2019). Compared with UCC, trace elements Co, As, La, and Nd are significantly enriched, while other elements are 385 depleted. Compared with the chemical elements in the Tenggeli and Badanjilin Deserts (Li, 2011), the Hexi 386 387 Corridor has lower SiO<sub>2</sub> content, similar K<sub>2</sub>O content, and lower contents of other trance elements.

Mineralogical and major- and trace-element geochemical analyses of aeolian sediments from the 388 Jinta-Gaotai area (area B in Fig. 2) in the middle Hexi Corridor have been carried out (Ferrat et al., 2011; 389 390 Wang and Wang, 2013; Wang et al., 2018; Zhang et al., 2020). These studies use the light/heavy mineral assemblages, the ratios of Na2O/Al2O3 to K2O/Al2O3 and SiO2/Al2O3, Ba/Sr to Rb/Sr, Rb/Sr to Ce/Sr, and the 391 392 composition of CaO and Cl to identify the provenance of aeolian sediments in the study area. Similar 393 mineralogical compositions (mica, guartz, illite, muscovite and albite) are found in dune sediments from 394 the Hexi Corridor and adjacent areas such as the Tenggeli and Badanjilin Deserts (Ferrat et al., 2011). This feature is also found by major and trace element analysis (Wang and Wang, 2013; Zhang et al., 2020), 395 396 indicating that the geochemical characteristics of aeolian sediments in the Hexi Corridor and its adjacent areas are also similar. Compared with the composition of the upper continental crust (UCC), dune 397 sediments in the Jinta-Gaotai area in the middle Hexi Corridor are also enriched in CaO (Fig. 6). Through 398 the methods of multi-dimensional scaling (MDS), principal component analysis (PCA) and regional 399 topography analysis, these studies suggest that the Hexi Corridor is not only the sediment sink of the Qilian 400 401 Mountains, but also the sediment sink of the Beishan Mountains. The sandy dunes from the Hexi Corridor 402 are similar in provenance to those from the Tenggelil, Badanjilin, and Kumtag Deserts (Zhang et al., 2020). 403

#### 404 4. Discussion

405

#### 4. 1. Physiochemical characteristics of sandy dunes in the Hexi Corridor 406

407

The above analytical results of geomorphological parameters indicate that in the dynamic process of 408 different dunes in the Hexi Corridor, the crescent-shaped dunes move the fastest, followed by the chains of 409 barchan dunes; only the top of the pyramid dunes wigwags, while the parabolic dunes and the longitudinal 410 dunes hardly move forward; the higher the height of the crescent-shaped dune (or the chain of barchan 411 dunes) is, the slower their movement is; on the contrary, the higher the height of the pyramid dunes is, the 412 413 faster they swing. Analysis also suggests that the moving speed of sandy dunes is positively correlated with the average wind speed of sandstorms. 414

Here, we compare the moving speed of sandy dunes in the Hexi Corridor with that in other desert 416 areas in northwest China. For example, the observation results of eight crescent-shaped dunes along the oil transportation highway in the Taklamakan Desert show that the moving speed of sandy dunes in 417 October 1991 $\sim$ 1992 is 4.81 $\sim$ 10.87 m, with an average of 7.29 m; the moving speed of sandy dunes in 418