# Peer review of "Formation and dynamics of sandy dunes in the inland areas of the Hexi Corridor"

_Solid Earth, 2020_

## Referee Comment (RC1) · Anonymous Referee #1 · 25 Sep 2020

General comments:

The paper of Bing-Qi Zhu focusses on formation and dynamic of changes of aeolian landforms (only sanddunes) and desertification in the Hexi Corridor in China. The methods used are satellite image interpretations, field investigations and observations, comprehensive evidences from geomorphological, aeolian‐physical, granulometrical and geochemical analysis to discuss the formation of dune landforms, the mechanism of desertification and their environmental implications in the Hexi Corridor.

The author concludes, that the Gobi area in the west Hexi Corridor is not the main source area of sandstorms in the middle and east of the corridor except north probably. In the past half century, the warming and humidification of local climate is the main cause of the reduction of sandstorms in the study area, and the Hexi Corridor has a

potential trend of anti‐desertification, which is mainly controlled by climate change but not by human activities. For the oases areas of the corridor, however, the effective measures to restrict desertification depend on human activities. Restriction of the decline of groundwater is the key to preventing desertification in oases. The abstract is a bit too long and should be shortened.

The focus on dune formation in the Hexi corridor in China is an interesting approach and could be of international interest - but this paper should be rejected in its current version for the following six main reasons:

(1) The paper discusses also desertification processes – which should appear in the title and in my opinion this is not the scope of this journal.

(2) The structure and data used for this paper is a bit confusing. The author used different topics, e.g. geomorphological parameters such as transport processes and dune movement, grain-size and geochemical data, meteorological data sometimes without clear connection between these different processes, data and the methods used. The manuscript shows in some parts a lot of details and methods, but at a closer look, quite a lot of things are missing (e.g. to the methods or the quality / origin of the data) or they are not appropriate. In part, this may result from the different data and sometimes confusing structure of the text, which is often not concise enough to see the main research question(s) and follow a line of arguments.

(3) The movements of sand including desertification processes were already mentioned and discussed decades ago, e.g. by Zhu et al. (1988). There is a long reference list including 32 papers in Chinese – but important international papers on desertification and sand movement are missing. There are more international papers focusing on desertification and dune movement worldwide and in China, which must be considered, e.g. the review on desertification in China from Wang et al. (2008). In addition, papers focusing on the aeolian sediments in the Hexi corridor (e.g. Nottebaum et al.) are not considered.

(4) The substantial conclusions by the author are: the Hexi Corridor has a potential trend of anti‐desertification, which is mainly controlled by climate change but not human activities. For the oasis of the corridor, however, the effective measures to restrict desertification depend on human activities. The first statement is based on the metrological data and not (enough) testified with other data, the second is not very new.

(5) What is not discussed and also important: The lowering of groundwater table in this region is due to the enlargement of the irrigation around the oasis especially after Second World War. This leads to reduction in river discharge and lowering of groundwater table and resulting in the drying of Gaxun Nur and adjacent lakes at the lower reaches of the Heihe River. This fact is already known and published since decades (e.g. Zhu Zhenda about 30 years ago). In addition, the lowering of discharge of groundwater table increased the salt content in the ground water and reduced areas with populous trees already in the 1980th of the last century

One other main point: It should be considered, why a reduction in groundwater caused dune movements. There is no relation between these two factors – except and maybe with flat sand sheets in the floodplains of larger rivers or close to lakes vegetated with trees. Quite a lot of trees in these areas were mainly destroyed in former times (before 1980) for firewood (see Zhu et al. 1988) and enhanced the desertification processes in sandy regions. However, the main and important factor for dune movement and desertification is grazing and overgrazing. This is not mentioned at all. Several of these aspects were described and discussed in numerous international papers. (6) Figures and language need to be improved (see some comments below).

Selected additional references (and references therein):

Zhenda, Zhu, Shu, Liu, Xinmin, Di (1988): Desertification and Rehabilitation in China. The International Centre for Education and Research on Desertification Control, Lanzhou.

Wang, X., Chen, F., Hasi, E., Li, J. (2008): Desertification in China: An assessment, Earth-Science Reviews 88: 188-206

Nottebaum, V., Lehmkuhl, F., Stauch, G., Lu, H., Yi, S. (2015): Late Quaternary aeolian sand deposition sustained by fluvial reworking and sediment supply in the Hexi Corridor – an example from northern Chinese drylands. Geomorphology 250: 113-127.

Specific comments:

Line 76-85: references needed

Line 165-168: Why this climate details at the end? Much better to climate line 143-142

Line 453-466: Heihe, grain-size discussed by Nottebaum et al.?

Line 525-526 . . .closely linked to regional land degradation. . .: Please specify, references?

Line 542-554 Please specify, references?

Line 617-622: already stated somewhere above

Line 624-625: Question should be in the introduction

Line 675-680: enlargement of irrigation not mentioned in detail

Fig. 1: Rivers and locations of the paper have to be mentioned

Fig.2: Please improve, difficult to read.

In addition, some of the other figures should be improved..
* * *

---

## Author Comment (AC1) · 25 Dec 2020

Comments from the reviewers

The interactive comment by the Anonymous Referee #1

General comments: The paper of Bing-Qi Zhu focuses on formation and dynamic of changes of aeolian landforms (only sand dunes) and desertification in the Hexi Corridor in China. The methods used are satellite image interpretations, field investigations and observations, comprehensive evidences from geomorphological, aeolian-physical, granulometrical and geochemical analysis to discuss the formation of dune landforms, the mechanism of desertification and their environmental implications in the Hexi Corridor. The author concludes that the Gobi area in the west Hexi Corridor is not the main

source area of sandstorms in the middle and east of the corridor except north proba-
bly. In the past half century, the warming and humidification of local climate is the main
cause of the reduction of sandstorms in the study area, and the Hexi Corridor has a
potential trend of anti-desertification, which is mainly controlled by climate change but
not by human activities. For the oases areas of the corridor, however, the effective
measures to restrict desertification depend on human activities. Restriction of the de-
cline of groundwater is the key to preventing desertification in oases. The abstract is a
bit too long and should be shortened. The focus on dune formation in the Hexi corri-
dor in China is an interesting approach and could be of international interest - but this
paper should be rejected in its current version for the following six main reasons: (1)
The paper discusses also desertification processes – which should appear in the title
and in my opinion this is not the scope of this journal. (2) The structure and data used
for this paper is a bit confusing. The author used different topics, e.g. geomorpho-
logical parameters such as transport processes and dune movement, grain-size and
geochemical data, meteorological data sometimes without clear connection between
these different processes, data and the methods used. The manuscript shows in some
parts a lot of details and methods, but at a closer look, quite a lot of things are missing
(e.g. to the methods or the quality / origin of the data) or they are not appropriate. In
part, this may result from the different data and sometimes confusing structure of the
text, which is often not concise enough to see the main research question(s) and follow
a line of arguments. (3) The movements of sand including desertification processes
were already mentioned and discussed decades ago, e.g. by Zhu et al. (1988). There
is a long reference list including 32 papers in Chinese – but important international
papers on desertification and sand movement are missing. There are more interna-
tional papers focusing on desertification and dune movement worldwide and in China,
which must be considered, e.g. the review on desertification in China from Wang et
al. (2008). In addition, papers focusing on the aeolian sediments in the Hexi corridor
(e.g. Nottebaum et al.) are not considered. (4) The substantial conclusions by the au-
thor are: the Hexi Corridor has a potential trend of anti-desertification, which is mainly

controlled by climate change but not human activities. For the oasis of the corridor, however, the effective measures to restrict desertification depend on human activities. The first statement is based on the metrological data and not (enough) testified with other data, the second is not very new. (5) What is not discussed and also important: The lowering of groundwater table in this region is due to the enlargement of the irrigation around the oasis especially after Second World War. This leads to reduction in river discharge and lowering of groundwater table and resulting in the drying of Gaxun Nur and adjacent lakes at the lower reaches of the Heihe River. This fact is already known and published since decades (e.g. Zhu Zhenda about 30 years ago). In addition, the lowering of discharge of groundwater table increased the salt content in the ground water and reduced areas with populous trees already in the 1980th of the last century. One other main point: It should be considered, why a reduction in groundwater caused dune movements. There is no relation between these two factors – except and maybe with flat sand sheets in the floodplains of larger rivers or close to lakes vegetated with trees. Quite a lot of trees in these areas were mainly destroyed in former times (before 1980) for firewood (see Zhu et al. 1988) and enhanced the desertification processes in sandy regions. However, the main and important factor for dune movement and desertification is grazing and overgrazing. This is not mentioned at all. Several of these aspects were described and discussed in numerous international papers. (6) Figures and language need to be improved (see some comments below). Selected additional references (and references therein): Zhenda, Zhu, Shu, Liu, Xinmin, Di (1988): Desertification and Rehabilitation in China. The International Centre for Education and Research on Desertification Control, Lanzhou. Wang, X., Chen, F., Hasi, E., Li, J. (2008): Desertification in China: An assessment, Earth-Science Reviews 88: 188-206. Nottebaum, V., Lehmkuhl, F., Stauch, G., Lu, H., Yi, S. (2015): Late Quaternary aeolian sand deposition sustained by fluvial reworking and sediment supply in the Hexi Corridor – an example from northern Chinese drylands. Geomorphology 250: 113-127. Specific comments: Line 76-85: references needed Line 165-168: Why this climate details at the end? Much better to climate line 143-142. Line 453-466: Heihe, grainsize discussed by Nottebaum et al.? Line 525-526 : : :closely linked to regional land degradation: : :: Please specify, references? Line 542-554 Please specify, references? Line 617-622: already stated somewhere above Line 624-625: Question should be in the introduction Line 675-680: enlargement of irrigation not mentioned in detail Fig. 1: Rivers and locations of the paper have to be mentioned Fig.2: Please improve, difficult to read. In addition, some of the other figures should be improved. Interactive comment on Solid Earth Discuss., https://doi.org/10.5194/se-2020-132, 2020.

 

The author's reply to the comments from reviewers

The author's reply to the interactive comments above by the Anonymous Referee #1

We are very grateful to the anonymous reviewer #1 for taking time and effort to read our submitted paper and for his/her detailed and constructive comments and suggestions on the manuscript. According to the review results of the anonymous Referee #1, we revised the manuscript point by point, and retained the trace of modification in the revised manuscript. The details of our one-by-one revision of the manuscript are listed and explained here, as followed below.

[The reviewer's comment] The abstract is a bit too long and should be shortened. [The author's replay] Yes, we agree with the reviewer and according to this comment, we revised the abstract section in the revised manuscript, especially shortened the length of the part.

[The reviewer's comment] (1) The paper discusses also desertification processes – which should appear in the title and in my opinion this is not the scope of this journal. [The author's replay] Yes, according to this comment of the reviewer, we revised the original text. As the reviewer suggested that it inappropriate to discuss the subject of desertification process in the paper submitted to Solid Earth, we reconsidered the topic discussed in our paper during the revision. We then deleted (or weakened) the content

about desertification issue in the study area in our revised manuscript. At the same time, according to this opinion of the reviewer, we also revised the title of the paper. Since the desertification part is no longer one of the topics discussed in this paper, we did not add the word "desertification" to the title of the revised paper. In addition, we would like to make a brief explanation on whether the desertification research falls within the scope of this journal (Solid Earth). As can be seen from the introduction of this journal's webpage, the scope of papers suitable to submit to the journal is that: Solid Earth (SE) is a not-for-profit journal that publishes multidisciplinary research on the composition, structure, and dynamics of the Earth from the surface to the deep interior at all spatial and temporal scales. It can be seen from this introduction that the subject/scope of this journal seems to include researches on surface processes of the earth such as the desertification issue, but it is not explicitly mentioned. We then consulted the paper bibliography that have been published in Solid Earth (SE) in the past, and found that some academic papers with the topic of desertification research had been published in the journal after peer review, including desertification researches both on the modern time scales and on the geological or historical time scales. Here we list some of them, shown as below in smaller font: Eskandari H, Borji M, Khosravi H, Mesbahzadeh T. Desertification of forest, range and desert in Tehran province, affected by climate change. Solid Earth, 2016, 7, 905-915. Costantini EAC, Branquinho C, Nunes A, Schwilch G, Stavi I, Valdecantos A, Zucca C. Soil indicators to assess the effectiveness of restoration strategies in dryland ecosystems. Solid Earth, 2016, 7, 397-414. Gui D, Xue J, Liu Y, Lei J, Zeng F. Should oasification be ignored when examining desertification in Northwest China? Solid Earth Discussions, 2017, doi:10.5194/se-2017-59. Li Y, Li Q, Luo G, Bai X, Wang Y, Wang S, Xie J, Yang G. Discussing the genesis of karst rocky desertification research based on the correlations between cropland and settlements in typical peak-cluster depressions. Solid Earth, 2016, 7, 741-750. Sadeghravesh MH, Khosravi H, Ghasemian S. Assessment of combating- desertification strategies using the linear assignment method. Solid Earth, 2016, 7, 673-683. Schwamborn G, Hartmann K, Wunnemann B, Rosler

W, Wefer-Roehl, Pross J, Schloffel M, Kobe F, Tarasov PE, Berke MA, Diekmann B. Sediment history mirrors Pleistocene aridification in the Gobi Desert (Ejina Basin, NW China). Solid Earth, 2020, 11, 1375-1398. Shoba P., Ramakrishnan SS. Modeling the contributing factors of desertification and evaluating their relationships to the soil degradation process through geomatic techniques. Solid Earth, 2016, 7, 341-354. Vieira RMSP, Tomasella J, Alvalá RCS, Sestini MF, Affonso AG, Rodriguez DA, Barbosa AA, Cunha APMA, Valles GF, Crepani E, de Oliveira SBP, de Souza MSB, Calil PM, de Carvalho MA, Valeriano DM, Campello FCB, Santana MO. Identifying areas susceptible to desertification in the Brazilian northeast. Solid Earth, 2015, 6, 347-360. Wang X, Hua T, Ma W. Responses of aeolian desertification to a range of climate scenarios in China. Solid Earth, 2016, 7, 959-964. Xie LW, Zhong J, Chen FF, Cao FX, Li JJ, Wu LC. Evaluation of soil fertility in the succession of karst rocky desertification using principal component analysis. Solid Earth, 2015, 6, 515-524. Xu EQ, Zhang HQ. Characterization and interaction of driving factors in karst rocky desertification: a case study from Changshun, China. Solid Earth, 2014, 5, 1329-1340. Zhang JY, Dai MH, Wang LC, Zeng CF, Su WC. The challenge and future of rocky desertification control in karst areas in southwest China. Solid Earth, 2016, 7, 83-91. However, in order to avoid ambiguity, and to respect the reviewer's opinion, we deleted the discussion of the subject of desertification in the study area in the revised manuscript. We then added the discussion on the geological processes (such as sedimentary processes and geochemical processes) responsible for the formation and evolution of aeolian dunes in the study area, according to the comments below and the new references/papers suggested by the reviewer.

[The reviewer's comment] (2) The structure and data used for this paper is a bit confusing. The author used different topics, e.g. geomorphological parameters such as transport processes and dune movement, grain-size and geochemical data, meteorological data sometimes without clear connection between these different processes, data and the methods used. The manuscript shows in some parts a lot of details and methods, but at a closer look, quite a lot of things are missing (e.g. to the methods

or the quality / origin of the data) or they are not appropriate. In part, this may result from the different data and sometimes confusing structure of the text, which is often not concise enough to see the main research question(s) and follow a line of arguments. [The author's replay] Yes, according to this comment of the reviewer, we have specially revised the section 2.2 "Methods and analytical data" of the original text in the revised manuscript. Due to the confusion in the structure, topic and logic of the original text, we specially emphasized or explained the analytical method used in this study, the reason and purpose of using the method, what kind of data we have obtained and what is the significance of the data obtained in the revised manuscript. Finally, we pay special attention to make this part consistent with the structures of the sections "3. Results" and "4. Discussion" in the late context and make them echo each other. In order to clearly show the revision results of this part in the revised manuscript, we present the full text of the revised section "2.2 Methods and analytical data" here for the reviewer and editor, as shown below in smaller font: "2. 2. Methods and analytical data Formation and characteristics of dune landforms are considered to be the result of a complex interplay between sediment properties and local preconditions (Nottebaum et al., 2015b). Although wind regime, surface condition, and sediment availability all control sand dune formation (Pye, 1995; Kocurek and Lancaster, 1999), geoscientists have mostly put emphasis on the wind regime. However, researches show that the formation of dune cannot be explained solely on wind because different dune types can form under the same wind regime in a given area (Rubin and Hesp, 2009; Lv et al., 2018). For example, in areas with little or no vegetation, dune formation depends strongly on wind regime and sediment availability (Baas and Nield, 2007; Wasson and Hyde, 1983), and both factors must be taken into account in studies on the formation and morphology of aeolian dunes; otherwise, modern dune morphology may lead to erroneous interpretations of the evolutionary history of dune environments (Rubin and Hesp, 2009). Therefore, sediment properties and local preconditions must be paid attention to. In detail, sediment properties are partly inherited from source materials, e.g. grain size and geochemical/mineralogical composition (e.g., Jahn et al., 2001; Prins

et al., 2007; Feng et al., 2011; Vandenberghe, 2013; Guan et al., 2013; Nottebaum et al., 2014, 2015b). Important local preconditions comprise surface and topographic properties, e.g. the geomorphologic setting, vegetation cover, and surface roughness on various spatial scales (Mason et al., 1999; Hugenholtz and Wolfe, 2010; Stauch et al., 2012, 2014; Nottebaum et al., 2014, 2015) and wind conditions (e.g. Pye, 1995; Lu et al., 2000; D. Sun et al., 2003; Kimura et al., 2009). Thus an insight into the potential relationship between dune-forming factors related to the wind regime, surface conditions and the available sediment sources within a dune system is needed. Based on this idea, in this study, our analysis methods and data mainly focus on the following aspects of dune landform: first, the data of morphological parameters varied with time in different dune types and dune units and the related acquisition method of these data are used to explore the characteristics of dynamic change and evolution of dune landforms in the study area. Secondly, the grain-size sedimentological data of sand dunes and their analytical methods are used to explore the characteristics of particles' mixing, potential transport mode and sedimentary maturity of aeolian sand in the study area, and to provide sedimentological clues about the wind forcing intensity and source area distance of dune sediments. Thirdly, the geochemical data and related analytical methods of dune sediments are used to determine the material sources of these detrital sediments in the study area. Fourthly, the meteorological, hydrological and climatic data of dune fields and the surrounding areas are used to explore the relationship between the dynamic evolution of dune landforms and environmental factors and its implications on desertification in the study area. These data and related analysis methods are described in detail below. For the study of dune geomorphology, the first method is to use the sample-quadrate survey procedure to measure the height and shape of typical high dunes in the field with a rangefinder, and the second is to measure the length, angle and width of the windward slope and downwind slope of each dune in the sample quadrate and between different quadrates along the local dominant wind direction by using rangefinder and remote sensing image scales (such as Google Earth scales, etc.), and then the comprehensive geomorphic data of sandy dunes in the region is obtained. In addition to the geomorphological data of sandy dunes themselves, landscape researchers will also use the sample-quadrate survey method to investigate the ecological parameters of vegetation cover in the selected sampling area. For both geomorphological and ecological surveys, sub-scale sample quadrates will be selected from the upper, middle and lower parts of the windward and leeward slopes of each dune. Three quadrates can be selected from the dune slope in the windward and downwind directions of each dune along the local prevailing wind direction and the size of each quadrate can be designed as 5m $\times$ 5m or smaller. In recent years, observation works have been carried out in different parts of the Hexi Corridor to investigate the different landform types of widespread sandy dunes at a geomorphic unit scale in the field (Chang et al., 2016a, 2017; Lang et al., 2017), including the crescent-shaped (barchan) dunes, chains of barchan dunes, pyramid-shaped dunes, parabolic dunes and longitudinal dunes belt. Based on time-series satellite remote sensing image data in different periods, the geomorphological parameters of these dunes are also obtained (Zhang and Dong, 2014; Chang et al., 2016a, 2017; Lang et al., 2017). In this study, the observation data and digital data were collected and sorted out systematically and corrected uniformly. Part of these comprehensive data of geomorphological parameters of sand dunes in the Hexi Corridor are shown in Tables 1 and 2. In addition to the above-mentioned intuitive survey and measurement of geomorphic parameter of sandy dunes, quantifying the structure of wind-blown sand flow and the movement rate of dunes is also the most direct and effective means to explain the dynamic change of dunes and their geomorphological evolution (Dong et al., 1998; Chen and Liu, 2011; He et al., 2012; Dong and Huang, 2013; J. Wang et al., 2013; Hu et al., 2016; Mao et al., 2016). Generally, there are two methods to study the moving velocity of sandy dunes, one is early positioning observation (MDCES, 1975; Dong et al., 1998; He et al., 2012; Shi et al., 2018) and the other is based on remote sensing images (Chen and Liu, 2011; Dong and Huang, 2013; Mao et al., 2016). Research works based on the both methods have been carried out in dune fields of the Hexi Corridor (MDCES, 1975; Chen and Liu, 2011; Chang et al., 2015, 2016a, 2017; Hu et al., 2016; Shi et al., 2018).

On this basis, this study integrates and organizes the different observation data of dune movement measurement in the Hexi Corridor. Parts of these data of dune movement measurement are shown in Table 3. Erodible clastic sediments as the material sources are the fundamental base for the formation of sedimentary landforms (Pettijohn et al., 1972; Taylor and McLennan, 1985). Therefore, identifying the composition, source, transport, and accumulation of wind-induced materials in an arid environment is a prerequisite for understanding the formation of dune landforms (Zhu et al., 1980, 1981a; Yang et al., 2012). Although sediment sections are usually used in many studies to observe the variability of sediment accumulation and sources in one location through time, in this study, the investigation approach of surface aeolian sediment samples at a regional scale is applied. The surface sample approach allows identification of distribution pathways by considering the continuous variety of geomorphological settings (Lehmkuhl, 1997; Kuster et al., 2006; Schettler et al., 2009; Nottebaum et al., 2014, 2015b; Zhu et al., 2014; Zhu and Yu, 2014). Therefore, this method offers an opportunity to analyze the spatial distribution and composition of surface aeolian sediments accumulated under variable conditions and from different perspectives (i.e. geomorphological setting, relief and vegetation). Grain size composition and distribution of aeolian sediment is an important indicator to understand the formation and development of sand dunes, because grain size distribution (GSD) analysis of sediments is an established tool to analyze the composition of sediments and to distinguish different transport, accumulation and potential remobilization processes (Folk and Ward, 1957; Friedman, 1961; Ashley, 1978; Bagnold and Barndorff‐Nielsen, 1980; McLaren and Bowles, 1985; D. Sun et al., 2002; Weltje and Prins, 2007; Qiang et al., 2007, 2010; Guan et al., 2013; Vandenberghe, 2013; Nottebaum et al., 2014; Zhu et al., 2014; Zhu and Yu, 2014). The depositional environment (aeolian, fluvial or lacustrine) and the movement types (creep, saltation or suspension) of sediments in the transportation process can be identified and distinguished by using grain size parameters of sand particles (Folk and Ward, 1957; Friedman 1961; Bagnold and Barndorff‐Nielsen, 1980; McLaren and Bowles, 1985; Nottebaum et al., 2014; Zhu et al., 2014). At present,

grain-size analysis of aeolian sediments and relevant detrital sediments, such as alluvial and proluvial fans, lacustrine deposits, fluvial deposits, has been widely carried out in the Hexi Corridor (Nottebaum et al., 2014, 2015b; Zhu and Yu, 2014; Zhu et al., 2014; Zhang and Dong, 2015; Zhang et al., 2016; Pan et al., 2019; Zhang et al., 2020). On this basis, this study systematically collects and organizes these granular data and records, which makes it possible to further conduct a comprehensive and comparative study on the dunes in the Hexi Corridor from a perspective of sedimentology. The analysis of major and trace elements, including rare earth elements, has become a reliable technique for detecting the source of desert sediments (Muhs et al., 1995, 1996; Pease et al., 1998; Honda and Shimizu, 1998; Wolfe et al., 2000; Pease and Tchakerian, 2003; Zimbelman and Williams, 2002; Muhs, 2004; Yang et al., 2007; Zhu and Yang, 2009; Jiang and Yang, 2019). The reason is that for aeolian sediments, the differences in compositions and distributions of rare earth elements and other trace elements in different samples/sub-fractions are largely controlled by the parent-rock compositions, because these elements only exist in specific minerals and are difficult to be lost during transportation (Pettijohn et al., 1972; Taylor and McLennan, 1985). In the Hexi Corridor, preliminary results have been achieved in the case studies of analyzing the elemental compositions of aeolian sediments using major- and trace-element geochemical methods (e.g., Schettler et al., 2009; X. Wang et al., 2010; Ren et al., 2014; Pan et al., 2019; Zhang et al., 2020), which provide basic data for this study to comprehensively identify the material sources of different dunes in the study area. The continuous data records of different meteorological parameters of local weather stations in the Hexi Corridor in the past half century, such as temperature, precipitation, relative humidity, wind speed, strong wind days and sandstorms days, will not only be the basis for this study to discuss the regional climate change under the background of global warming, but also the basis for exploring the response of regional landscape to climate change based on the statistical relationship between geomorphic parameters and climate parameters on a multi-decade time scale. Therefore, this study collects and uses the meteorological data of the Hexi Corridor for nearly half a century to analyze the regional climate change and its relationship with the dynamic changes of dune landforms."

[The reviewer's comment] (3) The movements of sand including desertification processes were already mentioned and discussed decades ago, e.g. by Zhu et al. (1988). There is a long reference list including 32 papers in Chinese – but important international papers on desertification and sand movement are missing. There are more international papers focusing on desertification and dune movement worldwide and in China, which must be considered, e.g. the review on desertification in China from Wang et al. (2008). In addition, papers focusing on the aeolian sediments in the Hexi corridor (e.g. Nottebaum et al.) are not considered.. [The author's replay] Yes, we agree with the reviewer's opinion. In recent decades, researches on desertification process and related wind-blown sand movement in northern China have been extensively carried out, and the relevant international literatures have been increasing greatly. In response to this comment of the reviewer, we have added some relevant and authoritative references to the revised manuscript, such as Wang et al., 2005(LDD); Wang et al., 2006a(GEC); Wang et al., 2008a(ESR); Notteaum et al., 2014(ESPL); Notteaum et al., 2015a(Geomorphology); Notteaum et al., 2015b(QI), etc, as listed below in smaller font. The quotations, reviews and discussions about the relevant contents inside these new international papers are inserted into the context all over the full text of the revised manuscript. As the text and content are abundant, so we will not repeat and list those words here. The detailed references newly cited are also not fully listed here because of the large number of them, but we can easily find them in the References section of the revised manuscript, because all revision trace are preserved in color. Nottebaum V, Lehmkuhl F, Stauch G, Hartman K, Wunnemann B, Schimpf S, Lu H. Regional grain size variations in aeolian sediments along the transition between Tibetan highlands and north‐western Chinese deserts – the influence of geomorphological settings on aeolian transport pathways. Earth Surface Processes and Landforms, 2014, 39, 1960-1978. Nottebaum V, Lehmkuhl F, Stauch G, Lu H, Yi S. Late Quaternary aeolian sand deposition sustained by fluvial reworking and sediment supply in the Hexi Corridor

- an example from northern Chinese drylands. Geomorphology, 2015a, 250, 113-127. Nottebaum V, Stauch G, Hartmann K, Zhang J, Lehmkuhl F. Unmixed loess grain size populations along the northern Qilian Shan (China): Relationships between geomorphologic, sedimentologic and climatic controls. Quaternary International, 2015b, 372, 151-166. Wang X, Chen F, Dong Z, Xia D. Evolution of the southern Mu Us Desert in North China over the past 50 years: an analysis using proxies of human activity and climate parameters. Land Degradation and Development, 2005, 16, 351-366. Wang X, Chen F, Dong Z. The relative role of climatic and human factors in desertification in semiarid China. Global Environmental Change, 2006a, 16, 48-57. Wang X, Wang T, Dong Z, Liu X, Qian Q. Nebkha development and its relationship to wind erosion and land degradation in semiarid northern China. Journal of Arid Environments, 2006b, 65, 129-141. Wang X, Chen F, Hasi E, Li J. Desertification in China: an assessment. Earth-Sciences Reviews, 2008a, 88, 188-206. Wang X, Xiao H, Li J, Qiang M, Su Z. Nebkha development and its relationship to environmental change in the Alaxa Plateau, China. Environmental Geology, 2008b, 56, 359-365. Wang X, Huang N, Dong Z, Zhang C. Mineral and trace element analysis in dustfall collected in the Hexi Corridor and its significance as an indicator of environmental changes. Environmental Earth Sciences, 2010, 60, 1-10. Wang X, Lang L, Hua T, Zhang C, Wang Z. Gravel cover of Gobi desert and its significance for wind erosion: an experimental study. Journal of Desert Research, 2013, 33(2), 313-319 (in Chinese with English abstract). Wang X, Hua T, Zhu B, Lang L, Zhang C. Geochemical characteristics of the fine-grained component of surficial deposits from dust source areas in northwestern China. Aeolian Research, 2018, 34, 18-26.

[The reviewer's comment] (4) The substantial conclusions by the author are: the Hexi Corridor has a potential trend of anti-desertification, which is mainly controlled by climate change but not human activities. For the oasis of the corridor, however, the effective measures to restrict desertification depend on human activities. The first statement is based on the metrological data and not (enough) testified with other data, the second is not very new. [The author's replay] Yes, we agree with the reviewer's opinion, and

combined with the reviewer's previous comments (1), we modified this part of the original text. Firstly we deleted the original content and weakened the discussion on the desertification issues in the study area. Secondly we added the discussion about the geological processes contributed to the formation and evolution of aeolian landforms in the study area. Please refer to the discussion section "4.1 to 4.4" of the revised manuscript for the specific revision results. Any traces of revision are preserved.

[The reviewer's comment] (5) What is not discussed and also important: The lowering of groundwater table in this region is due to the enlargement of the irrigation around the oasis especially after Second World War. This leads to reduction in river discharge and lowering of groundwater table and resulting in the drying of Gaxun Nur and adjacent lakes at the lower reaches of the Heihe River. This fact is already known and published since decades (e.g. Zhu Zhenda about 30 years ago). In addition, the lowering of discharge of groundwater table increased the salt content in the ground water and reduced areas with populous trees already in the 1980th of the last century. One other main point: It should be considered, why a reduction in groundwater caused dune movements. There is no relation between these two factors – except and maybe with flat sand sheets in the floodplains of larger rivers or close to lakes vegetated with trees. Quite a lot of trees in these areas were mainly destroyed in former times (before 1980) for firewood (see Zhu et al. 1988) and enhanced the desertification processes in sandy regions. However, the main and important factor for dune movement and desertification is grazing and overgrazing. This is not mentioned at all. Several of these aspects were described and discussed in numerous international papers.. [The author's replay] Yes, we agree with this opinion of the reviewer. The decline of groundwater is the main reason of regional desertification in the Hexi Corridor, and the decline of groundwater is due to the increase of upstream irrigation. In addition, the decline of groundwater level also leads to the increase of groundwater salinity and the degree of potential salinization risk in the study area. In response to this comment, and in combination with the previous comment (1) of the reviewer, we have revised the original discussion content of desertification issues in the Hexi Corridor. As stated above, we deleted and

weakened the discussion topic on desertification in the study area in the full text, and we added and strengthened the discussion topic about the geological process responsible for the formation and evolution of aeolian landforms in the study area. On the other hand, we actually referred to the relevant literatures and write some opinions of us towards the questions raised by the reviewer in this comment. Since the revised manuscript no longer discusses the topic of desertification, these new opinions of us did not occur in the revised manuscript, but we display this text here for review by the reviewer #1, as followed below in smaller font: "4. 5. Natural factors influencing land desertification in the Hexi Corridor Generally speaking, sandy dunes in the Hexi Corridor are dominated by the mobile dunes. In the early research work, the formation of these mobile dunes was considered to be caused by the destruction of vegetation previously fixed on the shrub dunes in the oasis; or because the gravel Gobi desert and related wind-erosion areas in the Hexi Corridor provided abundant sandy sediments, resulting in a wide aeolian-sand transportation and accumulation to form sandy dunes in the corridor area; or because that in the arid Hexi Corridor, the shallow and intermittent rivers with broad riverbeds changed their courses and the abandoned dried riverbeds were blown up by wind, causing accumulation of the fluvial sediments to form sandy dunes near the river banks (Zhu et al., 1980). The mechanisms above-mentioned for the formation of sandy dunes can explain the characteristics of the sporadic or sheet-like distribution of sandy dunes in the Hexi Corridor, especially the characteristics of sandy dunes intermittently distributed along the dried riverbeds by winding and zigzag patterns, and scattered on the gravel Gobi deserts at the edge of oases. Apart from the above natural causes, it is also believed that the formation of sandy dunes in the Hexi Corridor is not all controlled by the influence of natural factors, some dunes should be formed in historical periods and are the result of human activities (Zhu et al., 1980; Zhu and Chen, 1994; Zhu, 1999; Yang et al., 2004; Chang et al., 2005; Li, 2007). For example, ruins of the Han, Tang, and Ming Dynasties and sites of the Great Wall can also be found in some dune fields, such as the Shouchang ruins in the South Lake area of Dunhuang, the ruins in the dune fields of the Xicheng Post Station to the west of Zhangye,

and the sites of the Great Wall of Ming Dynasty in the dune fields in the Minqin area (Zhu et al., 1980; Zhu and Chen, 1994; Zhu, 1999), etc. Regarding the above desertification mechanisms, the wind-eroded fields in Gobi deserts and its deflation products in the Western Hexi Corridor (the upper windward area) are considered to be one of the reasons for the desertification of the middle and eastern Hexi Corridor, because the former may provide the material basis for the latter to form sandy dunes. However, is the existence of these western Gobi and wind erosion areas really the main cause of desertification in the Hexi Corridor? In order to solve this problem, researches have been carried out on the Gobi soils, wind-erosion landforms and the intensity and potential of the wind erosion process in the western Hexi Corridor (e.g., Zhu et al., 1980; Zhang et al., 2004; Qu et al., 2005; Wang et al., 2013; Yin et al., 2014, 2016; Zhang et al., 2016; An et al., 2019; Hu et al., 2020). From the perspective of geomorphology, wind-eroded lands are widely distributed in the western part of the Hexi Corridor. The result of surface wind erosion in these areas is the formation of long strip-shaped aeolian monadnocks and deflation hollows (aeolian depressions), which are roughly parallel to the wind direction. Generally, these wind-eroded landforms are about 1ï¡đ3 meters high, and a few are up to 5 meters (Zhu et al., 1980), which are roughly distributed along the lower reaches of the Shule River and the western part of the Dunhuang Oasis. Studies on the surface properties of gravel desert (Gobi) and its impact on wind erosion and dust emission in the west of the Hexi Corridor have shown that the gravel Gobi is very different from the sandy desert (dune field) because the grain-size composition of the surface sediment and the occurrence state (erodibility) of fine granular material are completely different (Zhang et al., 2004; Yin et al., 2014, 2016; Zhang et al., 2016; Hu et al., 2020). The surface of Gobi desert is composed of the gravel-, sand-, fine-sand- and clay-fraction particles with coarse- and fine-grained materials mixed together, while the surface of dune field is mainly composed of the medium-sand-fraction particles, with no or few coarser-grained gravels and finer-grained silt and clay particles (Zhang et al., 2016; An et al., 2019). A salt crust usually exists on the surface of Gobi desert in the west Hexi Corridor and thus the fine-grained particles are consolidated

due to the presence of salt cement, leading to a weak wind erodibility in Godi desert, while the surface of dunes is loose, making a strong erodibility in dune field (Zhang et al., 2016). Because the potential sediment availability (controlled by erodibility of surface fine-grained particles), the gravel coverage (surface roughness) and the average particle size of surface sediments are the main factors affecting dust emission (Pye, 1987; Gillette and Stockton, 1989; Raupach et al., 1993), the three factors will determine whether the Gobi desert in the western Hexi corridor has potential contribution to regional desertification in the Hexi Corridor. Based on field surveys and using the ImageJ software to process high-resolution image data, Zhang et al. (2016) estimated the gravel coverage and surface salt crust status in different Gobi areas in the west Hexi Corridor, determined the proportion of total surface sediment weight occupied by gravels (diameter > 2 mm), and analyzed the sedimentological grain-size distribution of different land surface sediments. The results of this work show that: (1) in the west Hexi Corridor, the gravel coverage of Gobi surface is moderate, with the gravel coverage mainly between 40% and 70% (average 52%, SD = $\pm$ 17%) (Zhang et al., 2016). The rate of gravel coverage in this level can produce the maximum aerodynamic roughness on the ground surface to prevent dust emission (Lyles and Tatarko, 1988; Wolfe and Nickling, 1996; Dong et al., 2002a, 2002b; Liu and Dong, 2003; Uno et al., 2006; Rostagno and Degorgue, 2011). (2) Most of the Gobi surface lands (75% of the total area) have formed salt crust. Only the areas with high sand-transport potential of wind in the northern Hexi Corridor and the edge of sandy deserts have no surface salt crust. (3) The content of erodible materials (sand, silt and clay) on the Gobi surface has a clear spatial distribution. Sediments of the Gobi surface are mostly the medium-sand and fine-sand particles (52.5% and 25.0%, respectively), while the silt and clay contents range from 9.8% to 40.1%, with most areas (about 73%) ranging from 10% to 30%. (4) In most Gobi regions, the potential transport of sand material is > 200 vector units, but 75% of these areas have solid soil crust on the ground surface (Zhang et al., 2016). Combined the above indexes, i.e. the proportion of fine-grained dust materials, the coverage rate of salt crust, and the potential transport of sand material, it is shown

that the high level of gravel coverage and surface crust rate in the west Heixi Corridor can effectively reduce the dust emission from the Gobi surface. Therefore, the Gobi area in the west Hexi Corridor is not the main source area of sandstorms occurred in the middle and east of the Hexi Corridor. The northern part of the Hexi Corridor may be the main source area of dust. Another potential indicator that can indicate the degree of modern desertification in the region comes from the meteorological parameter, such as the number of sandstorm days and strong wind days (Chang et al., 2019). Sandstorm refers to a windy and sandy weather phenomenon with the wind speed $\geq$ the sand-blowing wind speed and the horizontal visibility < 1,000 m. The Hexi Corridor is considered to be one of the areas with the most-frequent (heavy) sandstorm in northwest China (Zhu et al., 1994; Zhu, 1999). The results of meteorological data analysis from the Hexi Corridor show that since 1956, the number of local sandstorm days in the Hexi Corridor is about 11.20 day/year (Table 4), and the average number of strong wind days (wind speed >8 grade/day) is about 18.39 days/year (Table 4), but the number of local sandstorm days in the Hexi Corridor has shown a generally-decreasing trend (Fig. 7a), with a decline rate of 0.677 times/year (Chang et al., 2011). In addition, the frequency of sandstorms throughout northern China is also decreasing during the same period (Zhang and Ren, 2003; Li and Zhang, 2007). However, this is contrary to the situation of global sandstorms, because the number of times of global sandstorms is increasing in recent decades (Houghton et al., 2001; Ding, 2002). This indicates that the sandstorm process in the Hexi Corridor and even in the entire northern China does not respond to the global changes, revealing that the cause of desertification in the Hexi Corridor is different from other parts of the world. As shown in Fig. 7a, the number of sandstorm days in the Minqin area of the Hexi Corridor has shown a decreasing trend as a whole since 1956-2008, and there are three sub-trends during this period, i.e., both the frequency and number of sandstorms decreased rapidly (1956-1969), the frequency was high and stable (1971-1987), and the frequency was low and slowly decreased (1987-2008). However, with global warming, the temperatures in desert areas are generally increasing (Fig. 7b), and the frequency of sandstorms should have

increased, but why has the actual situation reduced? To answer this question, we need to analyze the climate change in arid northern China and the Hexi Corridor since this period. For nearly half a century, global warming has become a worldwide concern. In this context, how does the climate change and respond to global warming in the desert areas of northern China and the Hexi corridor? Researches on this issue have been carried out in arid regions of northern China and the Hexi Corridor (e.g. Ding, 2002; Sha et al., 2002; Chang et al., 2011, 2016b). Here, we briefly summarize the research results of climate change in the Hexi Corridor (the Minqin arean) in recent decades. (1) From 1961 to 2008, the rising rate of the annual average temperature in the Minqin area was higher than that of global temperature in the 20th century and that of China in the past 100 years. Among them, the temperature increase in February was the largest, with the monthly average temperature increasing by 3.01 ℃ (Fig. 7b). (2) From the 1980s to 1990s, the warmest in the world in the 20th century, the extreme maximum temperature in Minqin has increased significantly, while the extreme minimum temperature has decreased intermittently, and the instability of the extreme maximum and minimum temperatures has increased. The detailed results are as follows: the instability of the monthly average temperature in January and April increases; the isothermal date in February is 10.36 days earlier; the instability of the extreme maximum temperature in December and January increases; the variation coefficient of the extreme minimum temperature in May is as high as 287% (Fig. 7C). (3) During the period 1961-2008, the temperature in the Minqin area increased, the precipitation also shown an increasing trend, and the air humidity also increased significantly (Fig. 7d, 7f). (4) In the Minqin area, the instability of precipitation increased in January and the stability of annual precipitation increased (Fig. 7e). In general, as with the large-scale regional climate change in northern China, the problem of temperature instability should be more worthy of attention than the problem of temperature warming (Ding, 2002; Sha et al., 2002; Chang et al., 2011, 2016b). (5) The wind speed in the Minqin area continued to decrease during the period from 1961 to 2008, (Fig. 7g). (6) There is a significant negative correlation between the annual and seasonal distributions of

sandstorms and the relative humidity of air (Fig. 7h). It can be seen from the above that although the temperature in the Hexi Corridor has increased (in response to global warming), the precipitation has increased (in response to the enhancement of the Asian Summer Monsoon climate), the relative humidity of the air has increased and the wind speed has decreased. As a result of this environmental change, on the one hand, the dynamic force of dust-release process will be reduced (because of the decrease in wind speed), and on the other hand, the viscosity of the surface sediment particles will increase (because of the increase in humidity), which will reduce the frequency of dust storm events. Therefore, since 1961, the decreasing trend of sandstorm days in the Hexi Corridor is mainly due to the increase in the relative humidity of the air, that is, the warming and humidification of the local climate is one of the main reasons responsible for the reduction of sandstorms in the Hexi Corridor. In other words, the Hexi Corridor has a potential trend of reverse desertification in the past half century. The main influencing factor of desertification in the Hexi Corridor is climate change, that is, controlled by natural factors. 4. 6. Cause of man-made land desertification in the Hexi Corridor In addition to natural factors, some researchers believe that desertification in northern China is mostly driven by human factors. Zhu (1998) believed that in the past half century, excessive grazing (30.1%), excessive farming (26.9%), excessive logging (32.7%), improper use of water resources (9.6%), and inattention to environmental protection in the construction of industrial and mining transportation (0.7%) are the most important driving factors of desertification in the arid and semi-arid areas of northern China. These are all human-control factors under unnatural process, as shown in Table 6. However, some studies have analyzed the changing trends of various indicators about human activities and climate factors in semi-arid areas of northern China (such as the Hunshandake Sandy Land) from the 1950s to the early 21st century (as shown in Fig. 9). These indicators include the area of arable/cultivated land, the number of livestock and the size of the population, as well as precipitation, evaporation, temperature, sandstorm frequency and sand transport potential, etc. It was found that although human activities and population pressures have promoted desertification in the region,

but the effects of two climatic factors (sand transport potential and frequency of sand-storm) are much greater than those of the non-natural indicators (Wang et al., 2006). However, in the arid and extremely arid Hexi Corridor, the above-mentioned factors that cause desertification in the region are mainly water resources utilization (Wang et al., 2008). The evidence from digital image data and simulations shows that in a wide range of northern China, desertification usually occurs in arid and semi-arid areas with annual precipitation <450 mm and spring precipitation < 90 mm (Wang et al., 2008). The spatial pattern of desertification in these areas can be divided into two parts by drawing a line at about 100 E: the area east of the line is greatly affected by precipitation, wind activity and other climatic factors; in the area west of the line, desertification is affected by the same factors mentioned above, besides, land water resources (surface water flows and rivers originated from the melting water of glaciers, ice and snow) may play an important role in desertification or reverse desertification. In the Minqin Basin and in the northeast of the Hexi Corridor near the Alashan Plateau, the land degradation processes in these regions are proved to be mainly controlled by surface water resources and wind activities, while the impact of human activities such as reclamation on land degradation appear to have been overestimated in previous studies (Wang et al., 2008; Zhou et al., 2008). At a regional scale, significant declines in water inflows to the Hexi Corridor from 1949 to the present (Table 8) resulted in drying of lakes, decreases in groundwater levels (Tables 8, 9), and the expansion of areas of saline and alkaline lands and mobile sands (Zhang et al., 2008). From the 1970s to the present, most saline and alkaline and sandy lands and wadis were reclaimed in this region (Fig. 10), and irrigation of these lands depended mainly on groundwater because of the absence of surface water. However, the residence time of groundwater in this region is longer than 1,000 years (Shi et al. 1999), thus the groundwater levels will continue to decrease if there is no additional input of water (Zhang et al., 2008). The decreased water inflows into this region directly resulted in rapid desertification. From the mid-1970s to the mid-1990s, there was no significant change in the area of farmland (Fig. 10), but the areas of anchored and semianchored dunes decreased and

the area of semi-mobile dunes increased. This suggests that vegetation degradation on these dune surfaces accompanied the decrease in groundwater levels. However, the area of mobile dunes decreased after the mid-1990s as a result of a significant decrease in sand transport; this occurred because the period from the mid-1990s to the present had the lowest wind activity during the past 50 years in arid and semiarid China (Wang et al. 2006). However, different views have been expressed based on the study of regional groundwater change. Grace-GLDAS data and in situ observations have been used to evaluate groundwater dynamics and sustainability in the Hexi Corridor (e.g. Wang et al., 2020). Limited positive effects of the water management projects were detected on the groundwater system. The results indicated that (1) groundwater in the Hexi Corridor (HC) has experienced a general deterioration (except for a sudden and sharp increase observed around 2002) in both storage and sustainability, from $\triangle$GWS = 16.79 cm/year and SIGWS = 0.46 (1985–1990) to $\triangle$GWS = $-$28.96 cm/year and SIGWS = 0.008 (2007–2016); (2) the lowest value of groundwater sustainability in the HC appeared in the central and eastern regions (SIGWS = 0); (3) human activity was confirmed to be the dominant factor driving the processes of deterioration in groundwater sustainability in the HC, and during the research period, it is striking that relatively limited "positive" effects of the water management project were detected on the regional groundwater resource; this result indicates that damaged groundwater sustainability cannot be easily reversed unless a long-term management policy is implemented. In general, it can say that that groundwater in the Hexi area experienced a general deterioration in both storage and sustainability, and human activity was confirmed as the dominant factor driving the groundwater deterioration (Wang et al., 2020). For more than half a century, ecological studies also highlight that the ecological and water environment of the Hexi Corridor has changed greatly, such as: (1) The groundwater level in the Hexi Corridor has fallen and the regional water resources have decreased. For example, the 26 motorized wells located in the inner Minqin Oasis indicate that the groundwater level in the central part of the Minqin Oasis fell at a rate of 0.54m/y during the period 1985-2001 (Fig. 10); and the 7 motorized wells located at
the edge of the Minqin Oasis indicates that the groundwater level in the marginal area of the Minqin Oasis also decreased, with a drop rate of 0.56m/y from 1985 to 2017 (Fig. 10). (2) The area of natural sand-fixing forests in the Hexi Corridor has decreased. Due to the drop of groundwater level, the natural sand-fixing vegetation in the Minqin area has declined on a large scale, and the desert steppe has become sandy field. For example, in 1981, the area of natural sand-fixing forest in the Minqin area was 203,951 hm2, but by 2002, the area of natural sand-fix forest had decreased to 197,353 hm2 (Table 7); in the early 1980s, there were 373 hm2 of Populus euphratica forest in Minqin, but it has now disappeared (Table 7). We have developed generalized frameworks that represent the variations in precipitation, evaporation, sand-driving winds, water inflow, and proxies for human activity (population size and area of farmlands) over the past 50 years to assess their roles in desertification in the Minqin Oasis (Fig. 11). This framework suggest that although the vulnerability to desertification is a function of anthropogenic pressures, geomorphological processes, and climate change, in the Minqin Oasis and the surrounding region, water resource limitations have been more important. The above ecological problems tell us that in the Hexi Corridor, the reason for the continuous decrease of vegetation is not the lack of afforestation, nor the change of regional precipitation or relative humidity (although the climate in the Hexi Corridor has become more humid in the past half century, as illustrated in Fig. 6), but is subject to the changes of "effective water", such as groundwater and soil water (Fig. 10). In other words, effective moisture is the most restrictive factor for ecological sustainability in the desert environment. Therefore, the control of desertification in the hexi Corridor should be based on the principle of "centering on effective water balance". It is difficult to achieve the control of desertification without the overall direction of water-saving management and research on groundwater. Therefore, it can be said that the change of water environment is one of the leading factors restricting the change of the ecological environment from an oasis landscape toward a desertified landscape in the arid Hexi Corridor. The degradation of ecological environment in the lower reaches of the Shiyanghe River Basin in the Hexi Corridor, once again illustrates

this point. From a larger spatial scale, the arid and semi-arid areas in northern China are seriously deficient in water resources. How to combat desertification is the primary problem restricting the sustainable development of the region. Some scholars believe that the transfer of water from the outer basins (such as the Yellow River Basin and the Yangtze River Basin) can appropriately alleviate the serious water shortage and desertification in the inland area of the Hexi Corridor. However, is such a measure feasible? In view of the large-scale allocation of resources in arid areas, on the one hand, the transfer of water from the outer basin is not the fundamental solution to the problem, and it is likely to cause ecological problems in other watersheds, because almost every watershed under arid environment is short of water or potentially short of water. On the other hand, large-scale water transfer is also unrealistic. Due to the constraints of socio-economic conditions and financial resources in the Hexi Corridor, it is not feasible to transfer water from the outer basin at least in the near future. We believe that the fundamental way to solve the problem of water resources of the inland areas of the Hexi Corridor lies in the rational utilization of existing water resources. Furthermore, some studies believe that the inland areas of the Hexi Corridor are not an absolute "resource-based water shortage", but a combination of "resource-based water shortage" and "technics-based water shortage" coexist (Chang and Liu, 2003). It means that the utilization of water resources is unreasonable and the efficiency of utilization is not high. Comprehensive control is the only way to combat desertification in the interior of the Hexi Corridor."ãĂŚ

[The reviewer's comment] (6) Figures and language need to be improved (see some comments below). [The author's replay] Yes, in response to this comment of the reviewer, we have re-edited the original figures, re-checked the English language expression and errors in the full text, and made a thorough revision. Please refer to the main body of the revised draft for the Detailed revision processes and results can be found in the main body of the revised manuscript, in which any traces of our revisions are preserved.

[The reviewer's comment] Selected additional references (and references therein): (1) Zhenda, Zhu, Shu, Liu, Xinmin, Di (1988): Desertification and Rehabilitation in China. The International Centre for Education and Research on Desertification Control, Lanzhou. (2) Wang, X., Chen, F., Hasi, E., Li, J. (2008): Desertification in China: An assessment, Earth-Science Reviews 88: 188-206. (3) Nottebaum, V., Lehmkuhl, F., Stauch, G., Lu, H., Yi, S. (2015): Late Quaternary aeolian sand deposition sustained by fluvial reworking and sediment supply in the Hexi Corridor – an example from northern Chinese drylands. Geomorphology 250: 113-127. [The author's replay] Yes, according to this review opinion, we have added these documents to the revised manuscript. Not only that, we have also consulted and added other relevant documents. Due to the large number of these newly added international documents, they are not listed here. But we can easily find these new documents in the References section of the revised manuscript, as they are all preserved the traces of revision.

[The reviewer's comment] Specific comments: Line 76-85: references needed. [The author's replay] Yes, we agree with this opinion of the reviewer. However, in the revised manuscript, this part of the text and content has been modified greatly, so the reference documents originally needed are no longer necessary. But in the revised version, the corresponding references are added to the modified text and content. For details, please refer to the Introduction section "1." of the revised manuscript, as all traces of versions are preserved.

[The reviewer's comment] Line 165-168: Why this climate details at the end? Much better to climate line 143-142. [The author's replay] Yes, according to this opinion of the reviewer, we have made corresponding revision. We adjusted the information about climate to the Introduction of climate background in the revised manuscript. For details, please refer to the Background section "2.1." of the revised manuscript, as all traces of versions are preserved.

[The reviewer's comment] Line 453-466: Heihe, grainsize discussed by Nottebaum et al.?. [The author's replay] Yes, according to this opinion, we have referenced this

document detailedly in the revised manuscript and cited the results of the study of grain size sedimentology in this paper. For details, please refer to the Results and Discussion sections "3.3" and "4.2" of the revised manuscript.

[The reviewer's comment] Line 525-526: closely linked to regional land degradation: : :: Please specify, references? [The author's replay] In response to this comments of the reviewer, we discussed and answered in detail the relationship between the formation of shrub dune landforms and related regional land degradation in the oasis areas of the study area in the revised manuscript. For details, please refer to the Discussion section "4.4" of the revised manuscript.

[The reviewer's comment] Line 542-554 Please specify, references? [The author's replay] In response to the comment (1) put forward by the reviewer above, we have deleted this part of the discussion about the process and mechanism of desertification in the study area in the revised manuscript. Therefore, there is no special references needed to the original Line 617-622.

[The reviewer's comment] Line 617-622: already stated somewhere above. [The author's replay] In response to the comment (1) put forward by the reviewer above, we have deleted this part of the discussion about the process and mechanism of desertification in the study area in the revised manuscript. Therefore, there is no longer the text involving this issue of the original Line 617-622.

[The reviewer's comment] Line 624-625: Question should be in the introduction. [The author's replay] In response to the comment (1) put forward by the reviewer above, we have deleted this part of the discussion about the process and mechanism of desertification in the study area in the revised manuscript. Therefore, there is no longer the text involving this question.

[The reviewer's comment] Line 675-680: enlargement of irrigation not mentioned in detail. [The author's replay] On the one hand, in response to the comment (1) put forward by the reviewer above, we have deleted this part of the discussion about the
process and mechanism of desertification in the study area in the revised manuscript, so there is no longer the text involved irrigation and desertification here. But on the other hand, in the revised manuscript, we newly discussed the causes of formation of shrub dunes in oasis areas of the study region, involving the potential impact of irrigation on oasis dunes. We elaborates on this issue clearly. For details, please refer to the Discussion section "4.4" of the revised manuscript.

[The reviewer's comment] Fig. 1: Rivers and locations of the paper have to be mentioned. [The author's replay] Yes, in response to this comment of the reviewer, we introduced the background of the big rivers and other hydrological conditions in the study area in the Background section "2.1" of the revised manuscript, combined with Figs. 1 and 2. Please refer to Section 2.1 of the revised manuscript for details.

[The reviewer's comment] Fig.2: Please improve, difficult to read.. [The author's replay] Yes, in response to this comment of the reviewer, we redrew Fig.2 to make it clearer. Please refer to Fig. 2 of the revised manuscript for the detailed revision.

[The reviewer's comment] In addition, some of the other figures should be improved. [The author's replay] Yes, in response to this review opinion, we have also conducted a comprehensive inspection to all of figures in the paper and redrew them to make them more clear. Please refer to Figs. 1-8 of the revised manuscript for the detailed revision.

Please also note the supplement to this comment:
https://se.copernicus.org/preprints/se-2020-132/se-2020-132-AC1-supplement.pdf

―――――――――――――――――――――――――――

The map figure shows the geographical position of the Hexi Corridor in China, with labeled features including Badanjilin Desert, Kumutage Desert, Hexi Corridor, Minqin, Tenggeli Desert, and Gansu Province. Legend indicates Sandy desert, Minqin Oasis, and Shiyang River. An inset map shows China with Gansu and Beijing marked.

**Fig. 1.** Fig. 1 Geographical position of the Hexi Corridor in China

[Figure]

**Fig. 2.** Fig. 2 Geomorphological map of the Hexi Corridor (modified after T. Wang, 2003)

[Figure]

(a)

Barchan dunes

Movement speed (m/a)

dune 1, dune 2, dune 3, dune 4, dune 5, dune 6, dune 7, dune 8, dune 9, dune 10, dune 11, average

Year

(b)

Chains of barchan dunes

Movement speed (m/a)

dune 1, dune 2, dune 3, dune 4, dune 5, dune 6, average

Year

(c)

Movement speed (m/a) / Height (m)

Movement speed, Height

Barchan dunes

(d)

Movement speed (m/a) / Height (m)

Movement speed, Height

Chains of barchan dunes

(e)

Swing speed (m/a) / Height (m)

Movement speed, Height

Pyramid dunes

**Fig. 3.** Fig. 3 The moving speed and height of sandy dunes in the Hexi Corridor (modified after Chang et al., 2016a). (a) the moving speed of crescent-shaped (barchan) dunes; (b) the moving speed of chains of

[Figure]

300

250

200

150

100

50

0

Coefficient of variation (%)

Median grain size (μm)

Reworked silty loess
(RSIL)

Reworked sandy loess
(RSAL)

Stillwater silt
(SWS)

Slope deposit
(SD)

Silty loess (SIL)

Fluvial sand
(FS)

Sandy loess
(SAL)

Aeolian sand (AS)

AS
FS
SWS
SIL
RSIL
SAL
RSAL
SD

1                    10                    100                 1,000

**Fig. 4.** Fig. 4 The coefficient of variation (CV) plotted against the median grain size of classi-
fied surface sediment types in the Hexi Corridor (modified after Nottebaum et al., 2014). The
coefficient of var

[Figure]

**Fig. 5.** Fig. 5 The probability cumulative distribution curves of grain size of dune sediments in the Hexi Corridor (modified from Zhang and Dong, 2015)

[Figure]

The UCC-standardized distributions plots showing Sample/UCC for sand dunes:

(a) Badanjilin Desert (B) — samples 1–20

(b) Dune belt connecting Badanjilin Desert and Minqin Oasis (BM) — samples 21–41

(c) Dunes between the NW margin of Tenggeli Desert and the northeast of Minqin Oasis (TNE) — samples 65–76

(d) Dunes between Tenggeli Desert and the southeast of Minqin Oasis (TM) — samples 59–64

(e) Dunes between Tenggeli Desert and the south of Minqin Oasis (TSW) — samples 48–58

(f) Dunes in Minqin Oasis (M) — samples 42–47

Elements axis: Ba, Cr, Si$_2$O, Ni, Rb, Sr, V, Zr, Ti, Fe$_2$O$_3$, Al$_2$O$_3$, MgO, CaO, Na$_2$O, K$_2$O

**Fig. 6.** Fig. 6 The UCC-standardized distributions and compositions of major and trace elements of sand dunes in the Minqin Basin and its surrounding areas in the western Hexi Corridor (modified after Ren et a

[Figure]

**Fig. 7.** Fig. 7 Compositions of the major and trace elements and their UCC-standardized distributions of sandy dunes in the Jinta-Gaotai area of the middle Hexi Corridor (modified after Zhang et al., 2020)

**(a)** Prevailing wind

River

Arable lands

**(b)** Prevailing wind

Riverbed

Deserted
arable lands

Shrub dunes

**(c)** Prevailing wind

Shrub dunes

**(d)** Prevailing wind

Gobi desert surfaces

Mobile dunes

Shrub dunes

**Fig. 8.** Fig. 8 Formation and evolution of shrub dunes in the Hexi Corridor and neighboring areas (modified after X Wang et al., 2008b).

---

## Referee Comment (RC2) · Anonymous Referee #2 · 28 Dec 2020

I have enjoyed reading the manuscript, but, in its current form, the study is not fit for publication. This is mostly due to the uncertain intellectual contribution of the author. The manuscript does not represent a substantial contribution to scientific progress (substantial new concepts, ideas, methods, or data). It appears to be entirely based on previously published data sources. While the author gives due credit to these sources, it remains unclear where and to what extent findings in this manuscript transcend those from publications in which the data was originally conceived. The author does not give due credit to all literature, particularly western literature on the subject/from the region.

The manuscript is lengthy, and not succinctly written. The text would benefit from a rewrite including clarifications, and shortening. I will offer some comments (requests for clarification, see detailed comments) for the first paragraphs: the author should not

perceive these as a complete list, but an indication of how to also improve subsequent sections of the manuscript.

Coupling between text and figures is incomplete. Topographic terms mentioned (and referenced to particular figures even) are missing on maps, which hampers scrutiny of the research. Figures need improvements (see detailed comments).

Detailed comments (examples): L56: why should eolian sediments respond to global cimate change rather than regional or local climate changes??

L58: use units in conjunction with figures/numbers: 556,000 km2

L60: I'm not sure anything can cover a background.

L79-81: "...a dispute in people's understanding, such as "the theory of climate control", "the theory of tectonical/geomorphological control" and "the theory of groundwater control", geomorphological survey is essential to resolve this dispute.". Such mention requires referencing.

L82-84: "...where the dune landforms and desertification processes cover almost all the important information archives that understanding the earth system." Awkward phrasing: rewrite.

L87: The use of "especially wind and atmospheric circulation" appears highly redundant.

L89-90: "...China and some famous 90 steppes in history but are..." This is a highly unclear expression, and the use of "but" is questionable.

L91: you use both "Ka" and "myr" which is inconsistent. Moreover, kilo should be stated as "k".

L94: what global changes are referred to here?

L94: "...response to global changes. Therefore, it is of great significance..." there is

a logical gap between how dune morphology would reflect global climate change, and would merit the start of this sentence with "Therefore, it is of great significance...".

L97-98: "The formation and dynamics of sandy dunes in the world were observed and studied for the first time in the United States (Finkel, 1959) and the former Soviet Union (Znamenski, 1962) in the 1950s." These assertions make little sense, already in 1941 Bagnold published his landmark book, a Chinese(?) version of which you refer to as 1959: The Physics of Blown Sand and Desert Dunes. Weren't his observations from Libya?

L98-99: You then write "During this period, the famous desert physicist Bagnold..." except that his work originated from the 30's?

L104: Why the use of "However"? what's the contradiction?

L105: "...later development of refined and quantitative researches...". They already were quantitative.

L112: "... is considered to be the main source area and the engine area of...". Explain what the engine area means, and how it differs from being the source area?

L116-121: "It is unclear why this paper would go beyond the studies that have first reported the results on which this study leans."

L129: use unit.

L130: Alashan Plateau can't be found on a map.

L131: I wonder if "in" should be "to" and I believe you mean "separates" rather than "distributes"?

L132: Ulanbuhe Desert can't be found on a map.

L133: redundancy regarding location of Qilian.

L134-135: redundancy in the use of "such as" and etc.

L143: Add reference to substantiate the highly precise number.

L145: Add reference to substantiate reference to "a necessary place for the famous ancient Silk Road in China"

L147-148: you seem to imply that the onset of desertification is all anthropogenic: are there no climate drivers at all?

L149: "the combination"? unclear what is implied here

L152, 160: "west of Wushaoling and the east of Palaeo‐Yumenguan (Fig. 2)" none of the place names occur on Figure 2. Make sure not just a few but all the placenames occur in Figure 2.

L161-167: Unclear (final) paragraph: why is the Minqin Basin described in some detail, yet there is no illustration to help make a valid point and no references to back-up precise numbers?

L166-167: ". . .dunes are distributed on the northwestern edge of the oasis, i.e., the windward of sand‐transport winds in the oasis." Unclear description.

Figures: Apart from Fig. 1, there are no original figures in this manuscript, all are modified from previous sources.

Figure 1: Three rivers are mentioned in the text, only one of them is on the map. Please insert the others as well. It is unclear what relation the "Gansu" in the insert has to "Gansu Province" in the main map. Is the Hexi Corridor (different shade of grey) a part of the Gansu Province, or are both of them part of "Gansu" as per the inset map? The 40 N dashed line is wobbly rather than smooth.

Figure 2: This map cannot be appreciated sufficiently. Perhaps its readability can be improved if the three classes of dunes (mobile, semi-fixed, fixed) can be separated from each other with colors and moderate opacity. Is Gobi desert a fixed dune type? Certainly Oasis is not a fixed dune type? What doers (a)-(c) stand for? Please make

sure the rivers continue unbroken behind place names. Rivers are not part of the legend, neither are settlements. I can't see a good reason why all the coordinates have to be in half degrees: simplify by making it 39-51 N and 95-103 E. Qilian Mountains is repeated, although in one instance it is underlined, in the other it is not. In one instance it is Mountain and in the other Mountains. 100km should be 100 km.

Figure 3: Five mentions of "see Table 1 for geographical locations of the corresponding dune IDs" should be reduced to one. Also, dune numbering in Table 1 continues such that in panel b they should be numbered 12-17. Dune numbers on the x-axis in panels (d)-(e) need to be adjusted. Presumably, the values plotted in panels (c)-(e) are averages for 2006-2015; if so, this should be mentioned in the caption. Standardize the use of "pyramid" and "pyramidal" dunes. Why is panel (e) labelled "swing speed" and is it different from the others? If correct, change the caption phrasing.

Figure 4: Does the dune numbering follow Table 1? If so, make reference to Table 1. If not, explain in the caption what the numbering refers to. Also, this is indeed the same as Zhang and Dong safe for the colouring. I would additionally try to separate dunes 16 and 18 using filled symbols in one of them (use the same style as in Zhang and Dong).

Figure 5: spell out "UCC". To me, this looks like an exact copy of Ren et al. 2014; I'm really not sure what the modification is? This should be mentioned in the Figure caption. Make sure B, BM, TNE, etc... are defined in the figure caption.

Figure 6: This is exactly the same information as in referenced Zhang et al. (2020) but the panels have been rearranged. There is an omission from the original data that Co sample in panel (d) has an upper boundary of 8.5 according to the original figure. Explain UCC and 1950 (is this valid for all panels or merely panel (c))?

Figure 7: The panels have been rearranged relative to the figure caption. Is this the caption of the original publication? Panel (f) information (variation in average annual wind speed) does not exist?

Figure 8: is this "Depth to groundwater surface"?

Table 1: I'm not sure what "length of beaches" have to do with anything? There is no mention of beaches anywhere in the paper proper? It would be good to have a figure similar to Ren et al. 2014, Figure 1, with the locations of the dunes within the study area. This should be accomplished in a new Figure 1.

Table 2: why do the heights of Barchan dunes 7-11 not match those of Table 1 (but barchans 1-6 do!). Please explain "camponotus". Should "forward direction" be the same as "movement direction" in Table 1? They are again for dunes 1-6, but not 7-11? Again the term "beaches" is used. Why do these numbers not match those in Table 1. make sure to explain better in the table headers what differentiates the different tables (unless they are the same).

Table 4: change Weahter to Weather.

I am sorry I couldn't be more positive in my review of this submission, but I wish the author all the best in pursuing publication of this body of work from the Hexi Corridor, and I hope that my comments may help in that pursuit.